# Electronic inhomogeneity and phase fluctuation in one-unit-cell FeSe films

Dapeng Zhao [1,6], Wenqiang Cui [1,2,6], Yaowu Liu [2,6], Guanming Gong [2], Liguo Zhang [1], Guihao Jia [2], Yunyi Zang [1], Xiaopeng Hu [2], Ding Zhang [1,2,3], Yilin Wang [4] ✉, Wei Li [2,3], Shuaihua Ji [2,3], Lili Wang [2,3] ✉, Ke He [1,2,3], Xucun Ma [2,3] & Qi-Kun Xue [1,2,3,5] ✉

One-unit-cell FeSe films on SrTiO$_3$ substrates are of great interest owing to significantly enlarged pairing gaps characterized by two coherence peaks at ±10 meV and ±20 meV. In-situ transport measurement is desired to reveal novel properties. Here, we performed in-situ microscale electrical transport and combined scanning tunneling microscopy measurements on continuous one-unit-cell FeSe films with twin boundaries. We observed two spatially coexisting superconducting phases in domains and on boundaries, characterized by distinct superconducting gaps ($\Delta_1$~15 meV vs. $\Delta_2$~10 meV) and pairing temperatures ($T_{p1}$~52.0 K vs. $T_{p2}$~37.3 K), and correspondingly two-step nonlinear $V \sim I^{\alpha}$ behavior but a concurrent Berezinskii–Kosterlitz–Thouless (BKT)-like transition occurring at $T_{BKT}$~28.7 K. Moreover, the onset transition temperature $T_c^{onset}$~54 K and zero-resistivity temperature $T_c^{zero}$~31 K are consistent with $T_{p1}$ and $T_{BKT}$, respectively. Our results indicate the broadened superconducting transition in FeSe/SrTiO$_3$ is related to intrinsic electronic inhomogeneity due to distinct two-gap features and phase fluctuations of two-dimensional superconductivity.

Superconducting transition temperature ($T_c$) marks the inception of a macroscopic quantum phase-coherent paired state in fermionic systems. The paired fermionic system is described in a weak coupling Bardeen–Cooper–Schrieffer (BCS) limit or a strong coupling Bose–Einstein condensation (BEC) limit, scaled with a criterion $c \equiv 1/\left(k_F \xi_{pair}\right) \sim \frac{E_F^{1-6}}{\Delta}$. Here, $k_F$ is the Fermi momentum, $\xi_{pair}$ is the coherence length, $\Delta$ is the superconducting gap, and $E_F$ is the Fermi energy. Between these two limits, namely, the BCS-BEC crossover regime, incoherent fermion pairing occurs due to the Fermi surface instability at a pairing temperature $T_p$, forming the so-called pseudogap phase, while superfluidity of coherent pairs occurs at a lower temperature $T_c$. Given the characteristic superconducting gap magnitude $\Delta$ ~15–20 meV and the relatively small Fermi energy

$E_F$~56 meV, one-unit-cell (1 UC) FeSe on SrTiO$_3$(001) resides in the BCS-BEC crossover regime[5–7]. In coincidence with the crossover scenario, the hitherto observed zero-resistance temperature $T_c^{zero}$ of 20–30 K is much lower than the pairing temperature $T_p$ of 50–83 K, as disclosed by angle-resolved photoemission spectroscopy (ARPES) investigations[8–15].

The two-dimensional (2D) limit is a distinct feature of monolayer FeSe that relates closely to superfluidity. In strictly 2D, topological vortex-antivortex phase fluctuations give a Berezinskii–Kosterlitz–Thouless (BKT) transition[16–19]. The superfluidity disappears when the phase correlations change from a quasi-long range to short range order via the vortex-antivortex unbinding at a critical temperature $T_{BKT}$. Within the ideal BKT scenario, the $V(I)$

[1]Beijing Academy of Quantum Information Sciences, 100193 Beijing, China. [2]State Key Laboratory of Low-Dimensional Quantum Physics, Department of Physics, Tsinghua University, 100084 Beijing, China. [3]Frontier Science Center for Quantum Information, 100084 Beijing, China. [4]School of Integrated Circuits, Shandong Technology Center of Nanodevices and Integration, State Key Laboratory of Crystal Materials, Shandong University, Jinan 250100, China. [5]Department of Physics, Southern University of Science and Technology, Shenzhen 518055, China. [6]These authors contributed equally: Dapeng Zhao, Wenqiang Cui, Yaowu Liu. ✉e-mail: yilinwang@email.sdu.edu.cn; liliwang@mail.tsinghua.edu.cn; qkxue@mail.tsinghua.edu.cn

characteristics exhibit $V \sim I^\alpha$ power-law dependence, and the exponent $\alpha$ jumps discontinuously from 3 to 1 upon the temperature approaching $T_{BKT}$ from the lower side[20]. Indeed, previous $V \sim I^\alpha$ results indicate a BKT-like transition at a $T_{BKT}$ ($\alpha = 3$)-$T_c^{zero}$-23 K[9]. However, in sharp contrast to the full linear behavior within $1.1T_{BKT}$ for conventional superconductor films[21], the component $\alpha$ gradually decays to 1 until above $1.3T_{BKT}$, suggesting different mechanisms.

The coupling with the oxygen-deficient $TiO_2$ layer gives rise to the unique enhanced pairing in the 1 UC FeSe, characterized by double-gaps with two pairs of coherence peaks, namely one pair defined as inner gap at $\pm 10$ meV and the other pair as outer gap varying within $\pm 15$–$20$ meV[8,22]. The interfacial coupling strength and, thus, the pairing strength sensitively depends on the local interface stoichiometry, particularly the delicate competition between electron-doping from oxygen vacancies and hole-doping from extra Se adatoms[23,24]. Under the strong interface coupling limit, the 1 UC FeSe film consists of domains with diameters below hundred nanometers separated by unidirectional line defects, wherein the outer gap reaches at 20 meV in maximum. However, the gap magnitude decreases with reduced domain size and even vanishes in isolated domains with diameter below ~20 nm[25]. Besides the random and drastic variation of pairing gaps in the sub-micron scale, the pairing gaps exhibit lateral gradient reduction of ~2 meV per millimeter, resulting from the lateral oxygen gradient accumulation due to their spontaneous flow under the electric field[23]. Indeed, previous ARPES measurements from different groups reported varied gap magnitude $\Delta$ ~10–15 meV and pairing temperature $T_p$-50–83 K, due to the delicate interface coupling and strong spatial inhomogeneity.

In this work, we performed in-situ combined STM/STS and micron-scale electrical transport measurements and directly collected the spatially resolved pairing gaps and coherent pairing behavior in 1 UC FeSe. The micro-scale probes help to reduce the lateral pairing deviation due to the gradient distribution of oxygen vacancies. To reduce the sub-micro scale inhomogeneity, we prepared morphologically continuous 1 UC FeSe films on Nb-SrTiO$_3$(001) (0.7 wt %) under Se relatively rich condition, at the expense of weakened interface coupling featured by a maximum pairing gap of ~15 meV (for details, see Supplementary Figs. S1 and S2). Our results demonstrate the 1 UC FeSe with twin boundaries as a percolative system of two superconducting phases with $\Delta_1$-15 meV and $\Delta_2$-10 meV, exhibiting two-step nonlinear $V \sim I^\alpha$ behavior with contrasting $\alpha$-$T$ relations at low current and high current regimes.

## Results

### STM characterization of 1 UC FeSe

Figure 1a displays the topographic image of morphologically continuous 1 UC FeSe films following the terrace-step structure of the SrTiO$_3$(001) surface, with domains of several to tens of nanometers wide on terraces split by wavy bright boundaries. The zoom-in atomically resolved image displayed in Fig. 1b reveals structurally continuous Se-terminated (001) lattices over the whole surface with expanded Se(001) lattice ($3.82 \pm 0.05$ Å), compared with bulk FeSe. A detailed insight into the atomic structure around the boundary gives inversed lattice anisotropy in the adjacent domains, as marked by flipped white/yellow arrows ($a_0 > b_0$), and compressed in-plane Se(001) lattice ($3.70 \pm 0.05$ Å) in the $4a_0$–$6a_0$ wide boundary and lattice shifts upon merging, as illustrated by the staggered blue lines in the right panel. The above features indicate a twin boundary.

The contrasting lattice on boundaries induces varied pairing. Figure 1c shows the tunneling spectra taken across the boundaries at the locations labeled as colored dots in Fig. 1b. The superconducting gap reaches ~15 meV for regions away from the boundary, and reduces to ~10 meV in magnitude with significantly reduced coherence peaks and spectra weight loss due to pairing in 3 nm wide region spanned the border (the middle there curves). The 1 UC FeSe films with dense twin

boundaries can be regarded as a spatially percolative system consisting of two superconducting phases characterized by the respective superconducting gaps $\Delta_1$-15 meV and $\Delta_2$-10 meV.

Figure 1d, e shows the typical temperature-dependent d$I$/d$V$ spectra taken in domains and on boundaries, respectively. The superconducting gap magnitudes are $\Delta_1$-15 meV and $\Delta_2$-10 meV at 5.0 K. With increasing temperature, the gaps fill up, that is, coherence peaks maintain at $\pm \Delta_1$ and $\pm \Delta_2$, but the intensities gradually reduce and remain discernable till 32 K and 34 K, respectively, while the gaps remain visible until 50 K and above 34 K. No reliable spectrum is obtainable above 50 K due to thermal drift in our STM system. Figure 1f, g summarizes the corresponding temperature-dependent gap magnitude (blue squares) deduced from the BCS fitting (Supplementary Fig. S3b) and gap height (orange circles) defined as the difference between negative-bias coherence peak and zero bias conductance (exemplified by the arrowed dash in Fig. 1d). The blue and orange dashed lines correspond to the BCS fitting of gap magnitude (based on BCS gap function, see Supplementary Fig. S3b) and the linear fitting of gap height, respectively. The fittings on gap magnitude and gap height yield consistent $T_{p1}$-52.0 K and $T_{p2}$-37.3 K. Notably, the coherence peak of the superconducting gap ~15 meV vanishes at ~32 K (Fig. 1d and Supplementary Fig. S3a). We define this temperature as $T_{cp}$.

### In-situ micron-scale electrical transport measurement of 1 UC FeSe

Then we performed in-situ micron-scale electrical transport measurements to characterize the microscopic coherence. The silicon cantilevers coated with Au/Ti, as illustrated in Fig. 2a, b, directly contact the sample surface with an inter-probe separation of 5 μm. Under such a micro-scale soft contact, the current flows mainly through the thin surface layer, enabling the collection of electrical transport dominantly from the FeSe films rather than Nb-doped SrTiO$_3$ substrates (Supplementary Fig. S4). Figure 2c displays a typical temperature-dependent resistance ($R$-$T$) curve, with the temperature-dependent gap magnitude inserted for comparison. The resistance starts to decrease at $T_c^{onset} = 54.0$ K and drops completely to zero (defined as resistance within the instrumental resolution of $\pm 0.01$ Ω) at $T_c^{zero} = 31.0$ K. The crossing point between the linear extrapolation of the normal state and the superconducting transition is 43.8 K. Moreover, the zero-resistance state below $T_c^{zero} = 31$ K is further confirmed by the temperature-dependent $V$-$I$ behaviors shown in Fig. 2d. The $V$-$I$ curves at low temperatures show an apparent superconducting current plateau which becomes progressively shorter with increasing temperature and vanishes at approximately 31 K. The onset temperature $T_c^{onset}$-54.0 K and the zero-resistance temperature $T_c^{zero}$-31.0 K are consistent with $T_{p1}$ and $T_{cp}$, respectively, within experimental uncertainty. Note that under strong interface coupling, one can speculate a high $T_c^{onset}$ as well as the pairing temperature in 1 UC FeSe with the superconducting gap ~20 meV, which deserves further in-situ investigations (for details, see Supplementary Table S3).

We then check the $V(I)$ characteristics to gain deep insight into the transition region. Figure 3a summarizes the $V(I)$ curves on a double-logarithmic scale under temperatures varying from 19 to 35 K. Below 30 K, the $V(I)$ curves exhibit a two-step power-law $V \sim I^\alpha$ dependences, marked as regime H (high-current) and L (low-current), respectively, connected by a plateau marked in gray shadow. As the temperature increases, the plateau becomes progressively shorter until it vanishes at approximately 29 K. The $\alpha$-exponents are extracted from the power-law fittings in the two regimes respectively, as marked by short black dashed lines. Figure 3b presents detailed evolutions of the $\alpha$-exponents as a function of temperature. With decreasing temperature, the exponent $\alpha$ in the high current regime deviates from 1 at 30 K and approaches 3 at $T_H = 28.1$ K. At temperatures above 30 K, the $V(I)$ differs from the linear dependence and exhibits $\alpha < 1$. In contrast, the low current regime exhibits a broader transition and weaker exponent

variation: the exponent α deviates from 1 at a consistent temperature as regime H, approaches 3 at $T_L = 24.0$ K, and then fluctuates around 4 at lower temperatures. On the other hand, the observed $R(T)$ characteristics are consistent with a BKT transition. Close to $T_{BKT}$, the resistance follows the Halperin–Nelson equation $R(T) = R_0 \exp[-b(T/T_{BKT} - 1)^{-1/2}]$, where b is a material parameter[26]. As shown in Fig. 3c, the $(d\ln R/dT)^{-2/3}$ versus $T$ curve deviates from the linear relation at ~31 K and yields $T_{BKT}$ ~28.7 K, agreeing well with $T_H$.

The two-step power law $V \sim I^\alpha$ dependences evidence percolation via a network of superconducting puddles with different pairing strengths, echoing the two contrasting superconducting phases with $\Delta_1$ ~15 meV and $\Delta_2$ ~10 meV shown in Fig. 1. Within the BKT scenario, the α-exponent $\alpha(T) = 1 + \frac{\pi J_s(T)}{T}$, where $J_s$ is superfluid phase stiffness, $J_s = \hbar^2 n_s/4m = \hbar^2 c^2 d/16\pi e^2 \lambda^2$, with $n_s$ the 2D superfluid density, λ the penetration depth, and $d$ the film thickness[20]. In an ideal BKT transition, the α-exponent jumps discontinuously from 3 to 1 with the temperature from $T_{BKT}^-$ to $T_{BKT}^+$, correspondingly, $J_s(T)$ from $\frac{2}{\pi} T_{BKT}$ to 0. Beyond the ideal discontinuous α-exponent jump due to BKT transition, finite-size effect and inhomogeneity generally induce the broadening of the BKT transition[20]. Given the contrasting features in high-current and low-current regimes, the α-exponent transition in the former is closer to an ideal BKT transition. It is also supported by the $(d\ln R/dT)^{-2/3}$-$T$ analysis, giving consistent $T_{BKT}$ ~28.7 K with the $V$-$I^\alpha$ feature in the high-current regime. Notably, the $0.06 T_{BKT}$ transition is sharper than all the previous observations in 1 UC FeSe and SrTiO$_3$-based heterostructures[9,19]. Moreover, it is comparable to conventional 2D superconductors[21], indicative of improved homogeneity. With increased portions lost superconductivity under high current and high temperature, the current partially flows through the conductive substrate beneath the FeSe film, inducing trivial α < 1. The relatively broad transition in the low-current regime is likely due to the weak superconductivity in the weak superconducting puddles with much lower superfluid phase stiffness $J_s$.

Figure 4a shows the $R$-$T$ curves of 1-3 UC FeSe films, with $T_c^{onset}$ and $T_c^{zero}$ plotted as a function of film thickness. With increasing film thickness, the normal state resistivity decreases gradually, whereas the $T_c^{onset}$ significantly decreases with the second UC FeSe overlaid but varies little upon further FeSe deposition. Another more striking

feature is that the $T_c^{zero}$ remains at 31 K. This special thickness-dependent behavior demonstrates unique superconductivity in 1 UC FeSe rather than multilayer FeSe films, because the latter only contributes normal state conductivity but does not reduce superconducting fluctuation, which is the key factor to determine $T_{BKT}$ and thus $T_c^{zero}$. The significant decrease of $T_c^{onset}$ upon additional FeSe overlaid can be interpreted by weakened electron doping and strain effect in the monolayer FeSe, as disclosed by previous ARPES investigations[11]. The rich Se flux applied for tetragonal FeSe films deposition introduces hole doping to counteract the original electron doping. On the other hand, the overlayer FeSe films counteract the in-plane expansion enforced by the STO substrate.

## Discussion

Combined with previous ARPES characterization on the Fermi surface of 1 UC FeSe and the contrasting d$I$/d$V$ spectra on domains and boundaries, we present a comprehensive picture to depict the superconducting behavior of FeSe/SrTiO$_3$. The 1 UC FeSe with twin boundaries is a percolative system consisting of two superconducting phases, as illustrated in Fig. 4b. Compared with the phase with $\Delta_1$ ~15 meV in domains, the phase with $\Delta_2$ ~10 meV on boundaries exhibits blue-shifted kink of the valence band (for details, see Supplementary Fig. S5), corresponding to reduced electron pockets at M point[27], as illustrated in Fig. 4c, d.

Given the continuous 1 UC FeSe films and the minor content (<10%) of twin boundaries, as exemplified in Fig. 1a, the domains with $\Delta_1$ ~15 meV dominate the transport behavior, particularly under the superconducting state. This view is supported by the finding that $T_c^{zero}$ and $T_c^{onset}$ from global transport measurements are consistent with $T_{cp}$ and $T_{p1}$ from local spectroscopy measurements in domains, respectively. Moreover, the sharp BKT-like transition disclosed from the $V(I)$ behavior gives a $T_{BKT}$ close to $T_{cp}$. Right below $T_{BKT}$, zero resistivity is obtained, indicating a series of connected domains under the phase-coherent macroscopic quantum state. As the temperature reaches $T_{BKT}$, the proliferation of free topological vortices within the system leads to a sudden reduction of the superfluid density, concurrent with fluctuating coherence peaks and vanishing zero-resistivity. Since vortex dynamics is extremely disorder-sensitive in the 2D limit, the

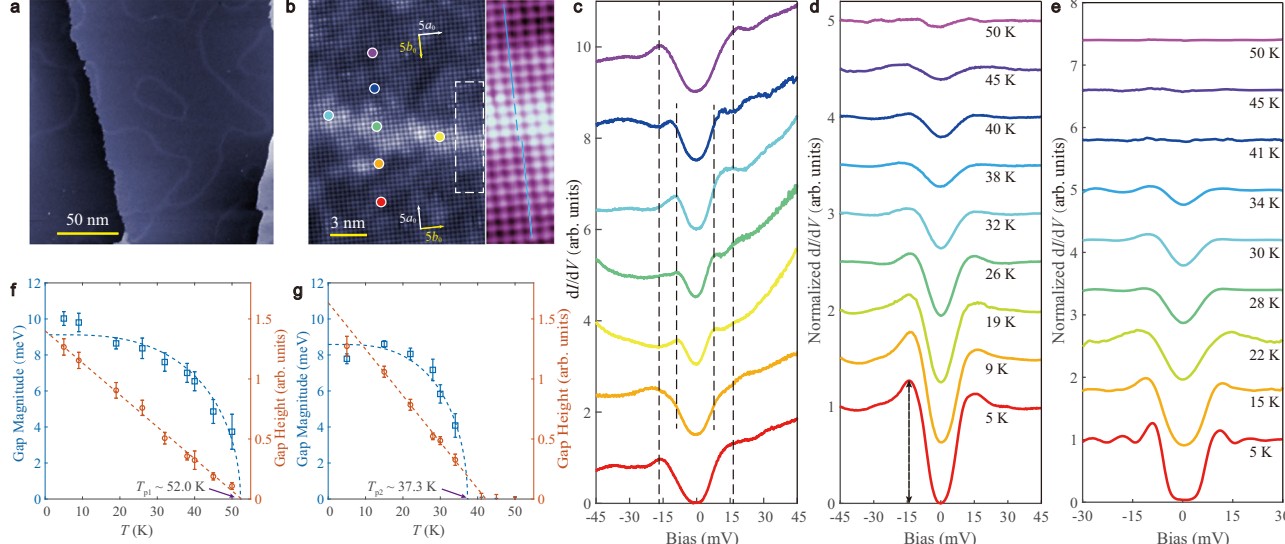

**Fig. 1 | STM characterization of 1 UC FeSe/SrTiO$_3$(001). a** Large-scale (sample bias $V_S = 1$ V, tunneling current $I_t = 10$ pA) and **b** zoom-in atomically resolved ($V_S = 0.1$ V, $I_t = 300$ pA) topographic images. **c** The d$I$/d$V$ tunneling spectra taken at the locations marked by colored dots in **b**. (set point: $V_S = 45$ mV, $I_t = 200$ pA, $\Delta V = 0.5$ mV). Temperature-dependent d$I$/d$V$ spectra taken **d** in domains (set point: $V_S = 45$ mV, $I_t = 200$ pA, $\Delta V = 0.5$ mV) and **e** on twin boundaries (set point: $V_S = 30$ mV, $I_t = 200$ pA, $\Delta V = 0.5$ mV). The spectra are normalized by dividing the raw spectra by their backgrounds obtained with the extrapolated method. **f, g** The gap magnitude and gap height of the d$I$/d$V$ spectra in **d** and **e** as a function of temperature, respectively. The error bars are from the s.d. of the fittings.

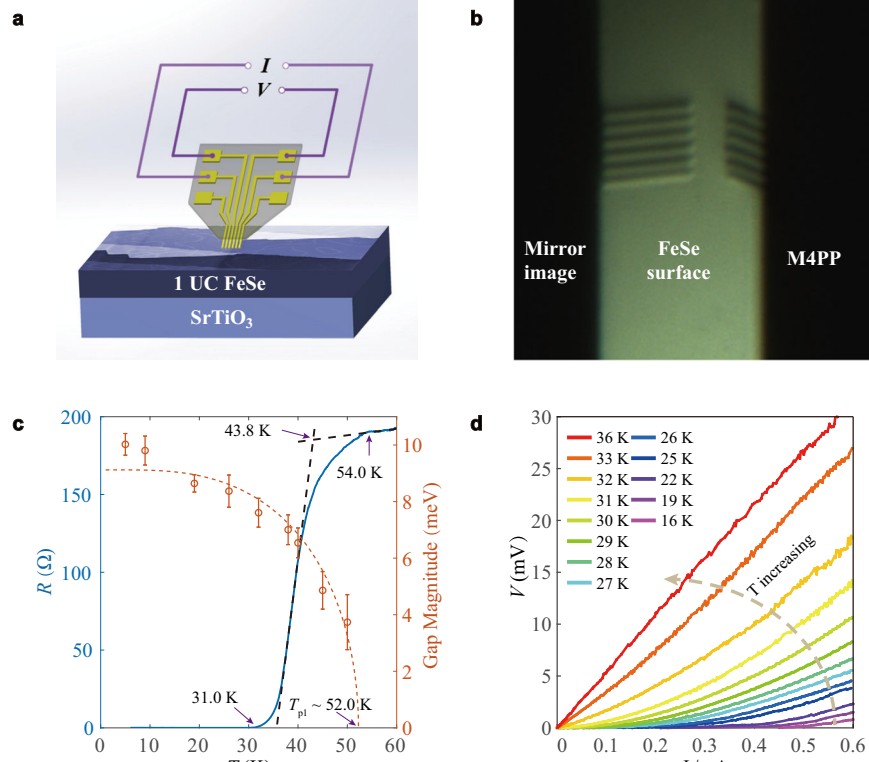

**Fig. 2 | In-situ micron-scale electrical transport measurements of 1 UC FeSe/SrTiO₃. a** Schematic of the measurement setup. **b** Optical image of the contacting process. The probes are spaced apart by 5 μm. The angle between probes and the sample surface is 30°, which allows soft contacts. **c** The resistance and the gap magnitude as a function of temperature, indicating the onset resistance-drop temperature agrees well with $T_{p1}$. **d** V-I curves at fixed temperatures from 16 K to 36 K, showing a superconductor-metal transition at ~31 K.

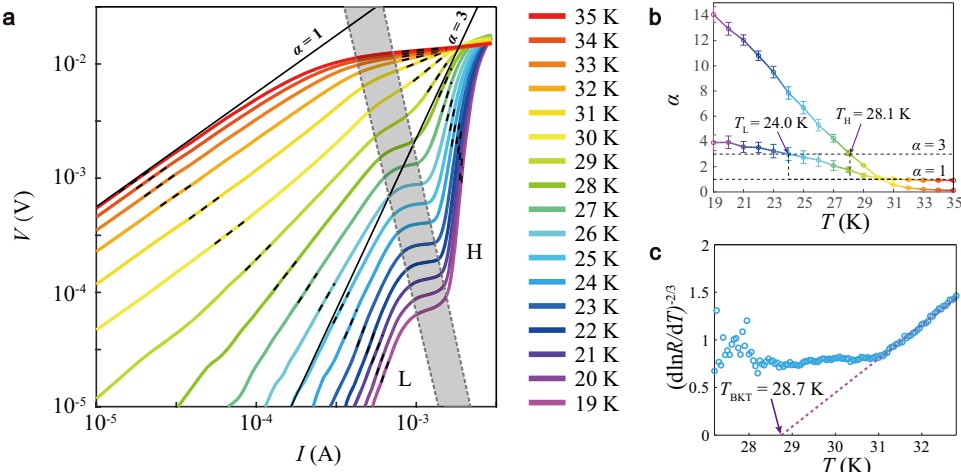

**Fig. 3 | BKT transition in FeSe/SrTiO₃. a** $V(I)$ characteristics from 19 K to 35 K on a double-logarithmic scale. Two solid lines correspond to $V \sim I$ and $V \sim I^3$ dependences, respectively. **b** Temperature evolution of the exponent α, extracted from the power-law fittings in regimes H and L in **a** giving $T_H = 28.1$ K and $T_L = 24.0$ K when α = 3. The error bars are from the s.d. of the fittings. **c** Temperature dependence of $(\mathrm{d}\ln R/\mathrm{d}T)^{-2/3}$. The purple dashed line shows the fitting to the Halperin–Nelson equation with $T_{BKT}$ ~28.7 K.

electronic inhomogeneity around the boundaries limits $T_{BKT}$ in 1 UC FeSe[20,28–31]. Upon either increased temperature or increased current, the superconductivity is destroyed locally around the boundaries with weak superfluid phase stiffness, and then in domains till the complete loss of superconductivity at $T_c^{onset}$, in coincidence with the gap closing temperature $T_{p1}$. The above picture is further supported by the two-step $V(I)$ behavior with a clear intermedia plateau, with the low-current

and high-current regimes dominated by the boundary and domain phases, respectively. In contrast, the $R$-$T$ behavior exhibits a wide transition in the temperature range between $T_{cp}$ and $T_{p1}$.

Moreover, a previous study on the highly two-dimensional bulk superconductor (TBA)$_x$FeSe, where the distance between adjacent FeSe layers is enlarged from ~5.5 Å in pristine FeSe to 15.5 Å by TBA⁺ intercalation, exhibits a sharper superconducting transition than 1 UC

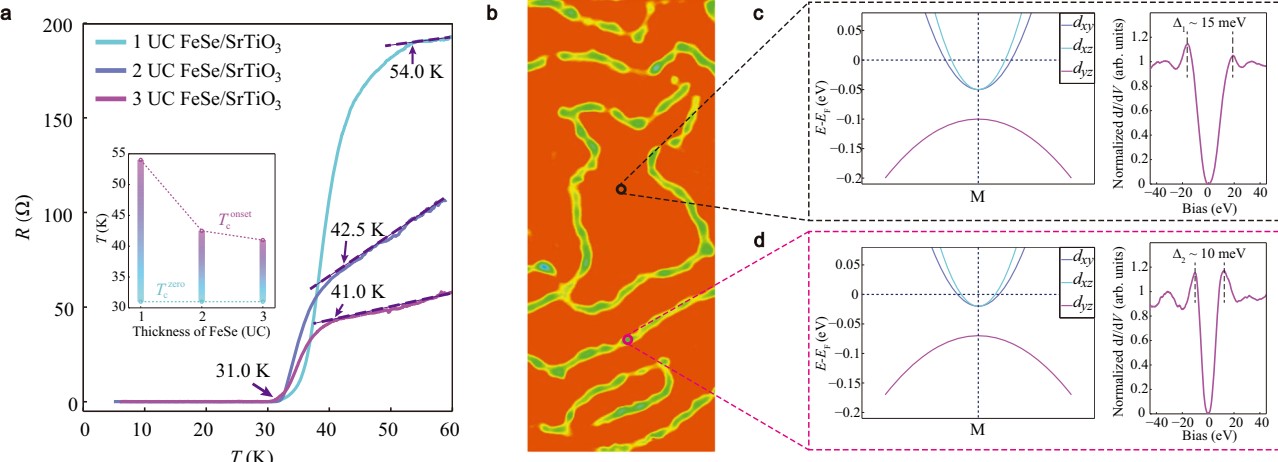

**Fig. 4 | Thickness variation and electronic inhomogeneity of FeSe/SrTiO₃. a** The resistance of 1-3 UC FeSe films as a function of temperature. The inset shows $T_c^{onset}$ and $T_c^{zero}$ as a function of film thickness. **b** Schematic illustrating the electronic inhomogeneity of FeSe/SrTiO₃. The orange area represents the superconducting

phase with $\Delta_1$ ~15 meV and the green area represents the superconducting phase with $\Delta_2$ ~10 meV. **c, d** Schematic of band structures at M point and representative d$I$/d$V$ spectra of two superconducting phases of 1 UC FeSe/SrTiO₃.

FeSe[32]. The onset temperature where the resistance starts to decrease is ~55 K, and $T_c^{zero}$ (~$T_{BKT}$) is ~43 K. The prominent difference between this system and 1 UC FeSe/SrTiO₃ is the lack of domains, which echoes that the electronic inhomogeneity around domain boundaries in 1 UC FeSe contributes to the broadening of superconducting transition.

It is worth noting that the 1 UC FeSe is characterized by a two-gap feature with robust inner gap at ±10 meV and sensitively varied outer gap within ±15–20 meV, while the $\Delta_2$ ~10 meV universally observed on twin boundaries agrees well with the inner gap value. Despite the missed two-gap feature in this work due to the reduced energy resolution[22] (increased oscillation under in-situ transport system combination), the unique two-step $V(I)$ features we presented here suggest electronic percolation system as well. The two-gap features and the pairing mechanism remain elusive. The results of this work may invoke more investigation and vision on the multiple-orbital properties in iron-based superconductivity[33–37].

In summary, we establish the direct comparison between spatially resolved spectroscopic probes on pairing and transport behavior of 1 UC FeSe/SrTiO₃ with twin domain boundaries by combining STM/STS and micron-scale electrical transport measurement technique under the same ultra-high vacuum. Our results reveal that the spatial electronic inhomogeneity and phase fluctuations cooperatively determine the transport behavior of FeSe/SrTiO₃. The former contribution is non-ignorable, but unfortunately underestimated in previous works. The in-situ combined spatially resolved spectroscopy and transport techniques presented in this work pave the way for eventually disclosing the pairing and coherence scenario in high-temperature superconductivity.

## Methods

Our experiments were conducted in a homemade ultra-high vacuum low-temperature STM - in-situ micron-scale electrical transport combined system, equipped with molecular beam epitaxy (MBE) and reflection high-energy electron diffraction (RHEED) for film preparation under the in-situ surface morphology monitor[38] (Supplementary Fig. S6). The base pressure is better than $1.0 \times 10^{-10}$ Torr. The Nb-doped SrTiO₃(001) (0.7 wt %) substrates were degassed at 600 °C for hours and then annealed at 1200 °C for 20 min to get atomically flat surfaces. FeSe films were then grown by co-evaporating high-purity Fe (99.995%) and Se (99.9999%) from standard Knudsen cells under Se-rich conditions as the substrate was heated to 420 °C. The growth rate is ~0.05 UC/min. At last, the samples were annealed at 460 °C for hours. The RHEED images (Supplementary Fig. S2) indicate a highly flat

SrTiO₃(001) surface and the epitaxial growth of single-crystalline FeSe films aligned with the SrTiO₃ substrates. All STM data were collected in a constant current mode at 5.0 K using polycrystalline PtIr tips calibrated on Ag films. The d$I$/d$V$ spectra were measured using a standard lock-in technique with a bias modulation of 0.5 mV at 973 Hz. In-situ electrical transport measurements based on the micro-four-point probe (M4PP) approach developed by Hasegawa et al. were performed under a built-in Pulse Delta measurement mode of Keithley Source-Meters 6221/2182 A with a pulsed current $I = 2$ μA applied[39,40] (Supplementary Fig. S7).

## Data availability

All data supporting the findings of this study are available within the paper and/or the Supplementary Information. Any additional requests for information of this study are available from the corresponding authors upon request.

## Code availability

The computer codes for STS fittings are provided with this paper.

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

## Acknowledgements

This work is supported by the National Natural Science Foundation of China (Grant Nos. 12304208, 52388201, 11427903, 92365108 and 12204043), the National Basic Research Program of China (Grant No. 2022YFA1403102), Innovation Program for Quantum Science and Technology (Grant Nos. 2023ZD0300500 and 2021ZD0302400), Beijing Natural Science Foundation (Grant No. 1222034), the Basic and Applied Basic Research Major Programme of Guangdong Province, China (Grant No. 2021B0301030003) and Jihua Laboratory (Project No. X210141TL210).

## Author contributions

D.-P.Z., Y.W., L.W. and Q.-K.X. designed and coordinated the experiments. D.-P.Z., W.C., Y.L., G.G., L.Z., G.J. and Y.Z. conducted film growth, STM/STS and transport measurements. X.H., D.Z., W.L., S.J., K.H. and X.M. participated in the data analysis. W.C. and Y.L. plotted the figures. D.-P.Z., L.W. and Q.-K.X. wrote the manuscript. All authors discussed the results and commented on the manuscript.

## Competing interests

The authors declare no competing interests.
