## [Peer Review File · Nature Communications]

Electronic inhomogeneity and phase fluctuation in one-unit-cell FeSe filmsREVIEWER COMMENTS

Reviewer #1 (Remarks to the Author):

This manuscript by Dapeng Zhao et al. presents a comprehensive study of monolayer FeSe using scanning tunneling microscopy (STM) and in situ four-probe measurements. The experiments are meticulously conducted, and the results are effectively presented. The authors highlight key findings, including the differences in the superconducting behavior between FeSe within the domains and at/near the domain boundaries.

One important aspect that warrants discussion is the claim of a "micro-scaled" setup, which is suggested to differ from "regular" sized transport measurements. The four-probe contacts used in this study are separated by 5 microns, while the domains themselves have sizes on the order of 50 nm. It is worth considering whether these measurements truly differ from those conducted on larger samples in terms of the domain effect. In essence, it may be possible that the smaller sample size merely reduces the inherent inhomogeneity, leading to sharper transitions and other observable differences. This aspect deserves further exploration and clarification in the manuscript. Regarding Figure 4 and the distinction between FeSe inside the domains and at the domain boundaries, an alternative explanation could be that these represent two distinct types of FeSe with different doping densities or superconducting transition temperatures and BKT transition temperatures. Such an interpretation could potentially account for the observed behaviors, including the complex BKT fittings in Figure 3. By considering the presence of two FeSe types with different T_{BKT} values, the intricate $V(I)$ behaviors could arise due to the involvement of vortex dynamics. While the manuscript mentions vortex and vortex-antivortex pair dynamics in different contexts, a more explicit physical picture is needed to fully understand these phenomena. There are a few minor points to address. Firstly, on page 4, the mention of "anisotropic in-plane lattice" lacks references or supporting results, and it would be beneficial to provide further information in this regard. Additionally, control experiments employing the micro-scaled probes on bare Nb-doped SrTiO₃ should be included to bolster the claim of predominantly collecting electrical transport from the FeSe films rather than the substrate. Furthermore, on page 5, the term "decreased lateral pairing deviation" should be further elaborated to provide a clearer understanding. Lastly, the claim regarding the 2 and 3-layer samples "not reducing superconducting fluctuation" requires additional clarification for better comprehension.

Reviewer #2 (Remarks to the Author):

Enhanced high- T_c superconductivity observed in monolayer FeSe/SrTiO₃ films has attracted much attention and research in recent years as an archetypical system for studying the impact of cooperative interfacial interactions on superconductors. Despite much effort, a complete understanding of the underlying phenomenology in this system remains elusive. A better understanding of the interplay between fluctuation effects, and nanoscale disorder is a critical hurdle in the study of 2D superconductivity more generally. This paper uses a combination of STM and in-situ micro to directly study this interplay.

From a technical perspective, the reliable micro-scale electrical transport measurements presented are impressive. In principle, combining transport with in situ STM is appealing, particularly for the study of air-sensitive 2D systems. The conclusion found, namely that the transport behavior in single-layer FeSe/STO is dominantly controlled by a combination of spatial inhomogeneity and phase fluctuation effects is likely correct. On the other hand, most of the results reported here are not particularly novel.

Qualitatively similar temperature-dependent STS have been previously reported even in the original discovery paper (Wang Qing-Yan et al 2012 Chinese Phys. Lett. 29 037402). The presence of nano-scale spatial inhomogeneity in the superconducting gap has been widely observed before

by many groups, as has the BKT-like behavior demonstrated by I-V characteristics (Fig. 3). Even the observations of gap structure variation across domain boundaries (Fig. 4b) is not novel, see: Fan et al. Nature Physics 11, 946–952 (2015).

The measured $R(T)$ behavior for monolayer films measured with micron-scale 4 point probe is very similar to that reported on macroscopic (mm sized) in situ measurements (PRX 11, 021054). However, the main text of the manuscript compares data to less-relevant ex-situ capped films, rather than this other in situ work. A direct comparison would be useful for clarifying that the relevant disorder impacting T_c must lie at the nano scale below the 5 micron probe spacing used here, but this was already reasonably well-understood from existing literature. The remaining data presented offers similarly few new insights.

The lack of fundamentally new results is the main issue with this work. In general, the technical and experimental aspects of the paper appear to be sound. However, I have some concerns about the authors' interpretation of some of the data. They claim to observe two distinct coexisting superconducting phases with different energy gaps, but it is not convincing from their data that the spatial gap distribution is so distinctly bimodal over large scale regions in the way they propose. This is also inconsistent with previous studies I am aware of such as Gong et al. PRB 100, 224504 (2019), which measured roughly normally-distributed spatial gap distributions. Considering that this is a critical aspect of their argument interpreting the I-V behavior as well, I would consider this a weakness in the work.

Additionally, the authors' interpretation of the temperature-dependent energy gap presented in Figure 1D and 1E is not particularly. In particular, a pure Dynes function fitting cannot be expected to give reliable results for the energy gap magnitude in the fluctuation-dominated (higher-temperature) regime. This is quite obvious when looking at the 40K data, for example, where the (residual) coherence peak position is nearly unchanged from the lower temperature data. I also notice that the gap values presented are also seemingly inconsistent between 1D and 1E; the coherence peak separation at 5K (Fig. 1D, red dI/dV curve) appears to be close to $2\Delta=30$ meV, but it is reported as approximately 10 meV in Fig. 1E. Can the authors explain this discrepancy? In any case, as good practice the fits used to generate the values in Fig 1E should be shown in the supplemental material (for all temperatures).

Reviewer #3 (Remarks to the Author):

The superconductivity of one unit cell FeSe has been a hot topic in recent years. Although there are a lot of ex-situ and in-situ transport measurements on FeSe films, consensus on its superconducting T_c has not been reached. In this manuscript, Zhao et al. combined STM/STS and in situ micro-scale transport measurements to study the one-unit-cell FeSe grown on SrTiO₃ substrate. Two main conclusions are as follows:

1. The zero resistance temperature is consistent with the coherence peak vanishing temperature, and the superconducting onset temperature is consistent with the gap closing temperature.
2. Two-step power law V-I dependence is observed. The authors explain this behavior as the percolation via a network of superconducting puddles with different superconducting gaps.

Currently, I cannot make a definitive recommendation as I have several questions and comments that the authors need to address.

I. To date, the superconducting T_c of FeSe on SrTiO₃ is controversial, and the proposed electronic inhomogeneity and phase fluctuations in this work could be a critical origin. However, the current study does not do a good enough job of discussing the main inconsistencies in previous literature.

1) This work is not the first combined work of STM and micro-scale electrical transport (the way of describing ref.25 and the first sentence of the summary should be revised). An early work, with some of the authors being the coauthors, has used quite similar techniques of 4PP to measure 1uc FeSe films, but the measured zero resistance temperature is above 100K (Ref.25). It's a bit confusing to the readers why the T_c varies so much on 1uc FeSe measured with similar

experimental techniques. The width between adjacent probes is even smaller in the current work (5 micrometers) than that in ref.25 (100 micrometers or 10 micrometers), which can reduce the effect of inhomogeneity. However, the measured T_c is much lower in this work, quite unexpected if electronic inhomogeneity is the main cause for the broadened superconducting transition. The discrepancy should be explicitly discussed.

2) Previous ARPES studies suggest pairing temperature around 65K. An early temperature-dependent STS work by the authors also suggests a pairing temperature of 68K (Zhang et al., *Phy. Rev. B* 89, 060506(2014)). Here the authors show the pairing temperature of 1uc FeSe is only 52K, much lower than reported, but consistent with the onset T_c . Is the reduced pairing temperature originated from the poorer sample quality? Do the authors expect the onset T_c of 68K on the optimized samples? In-situ transport measurements on the optimized samples with ~ 20 meV gap and two gap features should be shown. At least a discussion on the expected onset T_c for the optimized sample should be added.

II. The correspondence between the gap evolution and transport behavior cannot be directly observed from the raw data.

1) A third-order polynomial background is removed to analyze the gap size, while previous STS work does not show such a background. The authors should explain the origin of the background, its evolution with temperature, and how it would affect the gap determination.

2) According to the data in Fig.1d, the gap size seems constant below 45K, and the peak position always locates at about 15meV. The gap size of 45K seems even slightly larger than that of 40K. This is in contrast to the fitted gap size, which decreases with increasing temperature. The fitting using the Dynes function deviates from the data (Fig.S2). Is the poor fitting induced by the gap variation with temperature? The authors should show the fitted results at different temperatures.

3) The coherence peak can be hardly observed from the raw data (Fig.S2). Is it because of the poor sample quality? The definition of T_{cp} of 30K is based on the vanished coherence peak. However, one can still see a peak at around 15meV in the 32K or even 40K data. The authors need to explain how they define the coherence peak, so the readers can judge the existence by themselves. Besides, the peak structure might be related to the background subtraction. Can the author show the raw data and are there dramatic changes around 30K in the raw data?

III. For the micro-scale transport measurement, one doubt about Fig. 3 is how to prove that the low current regime and the high current regime are from the contributions of domain boundaries and domains, respectively. In the manuscript, the author claimed it's supported by the existence of an $\alpha=0$ plateau, which is not clear. Why would the coexistence of superconducting regions and non-superconducting regions induce a $\log V \sim \text{constant}$ region in IV curve? Because this is the main conclusion of this manuscript, more data should be shown to support this claim. For example, comparing $V(I)$ relation on the sample surface with different portions of domain boundaries and homogeneous domains would be helpful. Since the data in Fig.3 is from a highly homogeneous sample as the authors claimed, results from a more inhomogeneous sample should be shown and compared with Fig.3 to provide stronger evidence.

IV. The V-I curve in Fig.2d seems less smooth than the V-I curve in the double-logarithmic scale in Fig.3a. Are the data from different datasets or differently treated? Are they consistent if they are from different datasets?

Dear reviewers,

We sincerely thank all the reviewers for kindly handling our manuscript and providing scientific, rigorous and insightful comments and suggestions. We have considered all the comments and suggestions carefully and revised the manuscript accordingly. The responses to the reviewers' comments are listed as follows. The major modifications in the main text are highlighted in red color.

=====Authors' responses to the reviewers' comments=====

Response to the comment of Reviewer #1:

Reviewer #1 (Remarks):

This manuscript by Dapeng Zhao et al. presents a comprehensive study of monolayer FeSe using scanning tunneling microscopy (STM) and in situ four-probe measurements. The experiments are meticulously conducted, and the results are effectively presented. The authors highlight key findings, including the differences in the superconducting behavior between FeSe within the domains and at/near the domain boundaries.

Reply (R1-1): We thank the reviewer very much for the positive comment. Following his/her constructive suggestions, we have revised our manuscript accordingly.

One important aspect that warrants discussion is the claim of a "micro-scaled" setup, which is suggested to differ from "regular" sized transport measurements. The four-probe contacts used in this study are separated by 5 microns, while the domains themselves have sizes on the order of 50 nm. It is worth considering whether these measurements truly differ from those conducted on larger samples in terms of the domain effect. In essence, it may be possible that the smaller sample size merely reduces the inherent inhomogeneity, leading to sharper transitions and other observable differences. This aspect deserves further exploration and clarification in the manuscript.

Reply (R1-2): We thank the reviewer for the insightful comment. We agree with the reviewer's opinion. The measurement area of the micro-scale setup is still much larger than the domain diameter. However, compared with millimeter-scale transport measurement, the micro-scaled measurement helps to reduce the gradient gap magnitude in millimeter scale (*Phys. Rev. B* **100**, 224504 (2019)), which facilitates revealing the intrinsic nature of domain effect, such as the two-step nonlinear $V(I)$ behavior in this study.

Following this valuable comment, we have added more clarification in the third paragraph to explain the inherent inhomogeneity, and added "The micro-scale probes help to reduce the lateral pairing deviation due to the gradient distribution of oxygen vacancies" in the fourth paragraph to clarify the advantage of the micro-scale measurement in the revised manuscript.

Regarding Figure 4 and the distinction between FeSe inside the domains and at the domain boundaries, an alternative explanation could be that these represent two distinct types of FeSe with different doping densities or superconducting transition temperatures and BKT transition temperatures. Such an interpretation could potentially account for the observed behaviors, including the complex BKT fittings in Figure 3. By considering the presence of two FeSe types with different T_{BKT} values, the intricate $V(I)$ behaviors could arise due to the involvement of vortex dynamics. While the manuscript mentions vortex and vortex-antivortex pair dynamics in different contexts, a more explicit physical picture is needed to fully understand these phenomena.

Reply (R1-3): We thank the reviewer for the insightful comment. We agree with the reviewer's opinion that the observed $V(I)$ behaviors could be explained by the coexistence of two distinct types of FeSe with different doping densities, pairing strength and superconducting properties. Given the distinct two superconducting phases feature in 1 UC FeSe with twin boundaries and the advantage of micro-scale measurement as well as the bimodal Gaussian distribution of superconducting gap in Fig. S1d, we believe that the two FeSe types are the FeSe in the domains and the FeSe on the twin boundaries.

Regarding to the vortex and vortex-antivortex pair dynamics, considering the divergent α - T behavior in the high-current regime and the consistent $(d\ln R/dT)^{-2/3}$ - T analysis in Fig. 3, the V/I surge in the high-current regime is probably caused by the proliferation of free topological vortices, which is the well-known BKT transition. And given the existence of two types of FeSe (in domain and on twin boundaries), the V/I increase in the low-current regime is likely caused by the weak superconductivity of FeSe films on twin boundaries (compared with those in domains). As to the more detailed vortex dynamics of this two-fluid like system, we currently don't have a very clear physical picture due to the intricacies of the system.

Following this valuable comment, we have added more information to clarify the two-gap feature of 1 UC FeSe with twin boundaries and the advantage of micro-scale measurement to support our conclusion on Page 3 of the revised manuscript and in Fig. S1 of the revised supplementary information. Besides, we have added one paragraph on Page 8-9 of the revised manuscript to discuss the unique two-step $V(I)$ features.

There are a few minor points to address. Firstly, on page 4, the mention of "anisotropic in-plane lattice" lacks references or supporting results, and it would be beneficial to provide further information in this regard. Additionally, control experiments employing the micro-scaled probes on bare Nb-doped SrTiO₃ should be included to bolster the claim of predominantly collecting electrical transport from the FeSe films rather than the substrate. Furthermore, on page 5, the term "decreased lateral pairing deviation" should be further elaborated to provide a clearer understanding. Lastly, the claim regarding the 2 and 3-layer samples "not reducing superconducting fluctuation" requires additional clarification for better comprehension.

Reply (R1-4): We thank the reviewer for the nice suggestions. All the suggestions are accurate and valuable.

Firstly, we didn't explain "anisotropic in-plane lattice" clearly and have added "A detailed insight into the atomic structure around the boundary gives inversed lattice anisotropy in the adjacent domains, as marked by flipped white/yellow arrows ($a_0 > b_0$)" on Page 4 of the revised manuscript. The white/yellow arrows are marked in Fig. 1b.

Secondly, we have added the transport measurement result on bare Nb-doped SrTiO₃ substrate in Fig. S4 of the revised supplementary information, which shows no obvious kinks at transition temperatures of FeSe films. Due to the bad contact between M4PP and the substrate, direct contact without Au electrodes can't give any information. There, when measuring the resistance of SrTiO₃, Au electrodes are required to improve the contact.

Thirdly, the lateral pairing deviation is caused by the lateral gradient distribution of oxygen vacancies on SrTiO₃ surface along the heating current direction due to their spontaneous flow under the electric field (*Phys. Rev. B* **100**, 224504 (2019)). The micro-scale measurement could reduce the lateral pairing deviation. However, as the second reviewer points out, it is probably not appropriate to compare our results with previous *ex situ* studies. Thus, we have deleted this part as well as the mention of "decreased lateral pairing deviation".

Lastly, T_c^{zero} is consistent with T_{BKT} , which is influenced by phase fluctuations. In 1-3 UC FeSe films, T_c^{zero} remains at 31 K, indicating that superconducting fluctuation doesn't be reduced with increasing film thickness and the superconductivity only occurs in 1 UC FeSe. We have added more explanation accordingly in the revised manuscript, "because the latter only contributes normal state conductivity but does not reduce superconducting fluctuation, which is the key factor to determine T_{BKT} and thus T_c^{zero} ".

Reviewer #2 (Remarks):

Enhanced high-T_c superconductivity observed in monolayer FeSe/SrTiO₃ films has attracted much attention and research in recent years as an archetypical system for studying the impact of cooperative interfacial interactions on superconductors. Despite much effort, a complete understanding of the underlying phenomenology in this system remains elusive. A better understanding of the interplay between fluctuation effects, and nanoscale disorder is a critical hurdle in the study of 2D superconductivity more generally. This paper uses a combination of STM and in-situ micro to directly study this interplay.

Reply (R2-1): We thank the reviewer very much for reviewing our manuscript. Following his/her constructive comments and suggestions, we have revised our manuscript accordingly.

From a technical perspective, the reliable micro-scale electrical transport measurements presented are impressive. In principle, combining transport with in situ STM is appealing, particularly for the study of air-sensitive 2D systems. The conclusion found, namely that the transport behavior in single-layer FeSe/STO is dominantly controlled by a combination of spatial inhomogeneity and phase fluctuation effects is likely correct. On the other hand, most of the results reported here are not particularly novel.

Reply (R2-2): We thank the reviewer for the valuable comment. We are glad that the reviewer endorses the method and conclusion of this study. The superconducting properties of 1 UC FeSe are closely related with the sample quality, such as the density of line defects or twin boundaries. Different sample quality makes meaningful comparisons of previous studies difficult, impeding understanding the underlying superconducting mechanism. Therefore, we determine to set up an *in situ* system to combine STM/STS and transport measurements. The data presented in the manuscript are not aimed to repeat the previous studies, but to show the sample status to the readers. Our key findings are as pointed out by the third reviewer:

1. The zero resistance temperature is consistent with the BKT transition temperature, which is also close to coherence peak vanishing temperature, and the superconducting onset temperature is consistent with the gap closing temperature.
2. Two-step power law V - I dependence is observed. This behavior could be explained as the percolation via a network of superconducting puddles with two superconducting phases.

In addition, we have added the temperature evolution of dI/dV spectra taken on twin boundaries in Fig. 1 of the revised manuscript, which are also new findings.

Qualitatively similar temperature-dependent STS have been previously reported even in the original discovery paper (Wang Qing-Yan et al 2012 Chinese Phys. Lett. 29 037402). The presence of nano-scale spatial inhomogeneity in the superconducting gap has been widely observed before by many groups, as has the BKT-like behavior demonstrated by I-V characteristics (Fig. 3). Even the observations of gap structure variation across domain boundaries (Fig. 4b) is not novel, see: Fan et al. Nature Physics 11, 946–952 (2015).

Reply (R2-3): We acknowledge the above similar findings have been reported previously. However, different sample quality gives different measurement results. We think it is crucial to know the sample status down to atomic scale for understanding its superconducting behaviors. *In situ* combination of STM/STS and transport measurements facilitates establishing the direct correlation between pairing and transport behavior of 1 UC FeSe without the confusion of sample quality, which is what we want to do in this study.

Furthermore, we have made it clear that there are two types of 1 UC FeSe films in the introduction of the revised manuscript and Fig. S1 of the revised supplementary

information. Under strong interface coupling limit, the domains are separated by unidirectional line defects of Fe vacancies, around which the superconductivity gets suppressed. The gap magnitude decreases with reduced domain size and even vanishes if the diameter of a domain is smaller than ~ 20 nm (*Phys. Rev. Mater.* **6**, 064803 (2022)). So the superconducting gaps are influenced by the domain diameter, resulting in an approximate Gaussian distribution (*Phys. Rev. B* **100**, 224504 (2019)).

Under Se relatively rich condition, the structurally continuous 1 UC FeSe film can be prepared, which consists of bright twin boundaries, at the expense of weakened interface coupling. Twin boundaries are also FeSe films with compressed lattice (3.70 ± 0.05 Å), across which the lattices of adjacent domains are orientated perpendicularly with an $a/2$ lattice shift, as shown in Fig. 1b. Without abundant Fe vacancies, the 1 UC FeSe film with twin boundaries exhibits two typical superconducting gaps, namely ~ 15 meV in domains and ~ 10 meV on boundaries. In light of that, this kind of 1 UC FeSe film can be considered as a network of superconducting puddles with two superconducting phases, as evidenced by the bimodal Gaussian distribution of superconducting gap in Fig. S1d.

As the composition and the superconducting gap distribution of two kinds of FeSe films are different, we should study them separately. However, nearly all the previous literatures focused on FeSe films with line defects boundaries, which have larger superconducting gaps. To our knowledge, only *Nat. Phys.* **11**, 946-952 (2015) has made clear that the sample is composed of twin boundaries, but lacks of transport measurements. Thus, FeSe films with twin boundaries need further research.

The measured $R(T)$ behavior for monolayer films measured with micron-scale 4 point probe is very similar to that reported on macroscopic (mm sized) in situ measurements (PRX **11**, 021054). However, the main text of the manuscript compares data to less-relevant ex-situ capped films, rather than this other in situ work. A direct comparison would be useful for clarifying that the relevant disorder impacting T_c must lie at the nano scale below the 5 micron probe spacing used here, but this was already reasonably well-understood from existing literature. The remaining data presented offers similarly few new insights.

Reply (R2-4): We thank the reviewer for the very helpful comment. We agree with the reviewer's opinion that it is probably not appropriate to compare with previous *ex situ* studies and a slight T_c^{zero} increase is well-understood under micro-scale measurement compared with *Phys. Rev. X* **11**, 021052 (2021).

Following this valuable comment, we have deleted this comparison in the revised manuscript.

The lack of fundamentally new results is the main issue with this work. In general, the technical and experimental aspects of the paper appear to be sound. However, I have some concerns about the authors' interpretation of some of the data. They claim to observe two distinct coexisting superconducting phases with different energy gaps, but

it is not convincing from their data that the spatial gap distribution is so distinctly bimodal over large scale regions in the way they propose. This is also inconsistent with previous studies I am aware of such as Gong et al. PRB 100, 224504 (2019), which measured roughly normally-distributed spatial gap distributions. Considering that this is a critical aspect of their argument interpreting the I-V behavior as well, I would consider this a weakness in the work.

Reply (R2-5): We thank the reviewer for the valuable comment. The reviewer points out two main issues: (1) lack of fundamentally new results; (2) the spatial gap distribution is inconsistent with previous studies. This is because we haven't introduced the background clearly. Accordingly, we have added more information to clarify the difference between two types of 1 UC FeSe on Page 3 of the revised manuscript and Page 2 of the revised supplementary information.

For the first issue, we have explained and highlighted the key findings in the reply (R2-2). For the second issue, we have explained in the reply (R2-3). The FeSe films studied in *Phys. Rev. B* **100**, 224504 (2019) with line defects domain boundaries were different from this study. Affected by dense line defects of Fe vacancies, the spatial gap distribution is roughly normally-distributed. The FeSe films with twin domain boundaries, studied in this work, are structurally continuous and can be regarded as a percolation system with two superconducting phases, as evidenced by the bimodal Gaussian distribution in Fig. S1d. Then the $V-I$ behavior can be observed.

Additionally, the authors' interpretation of the temperature-dependent energy gap presented in Figure 1D and 1E is not particularly. In particular, a pure Dynes function fitting cannot be expected to give reliable results for the energy gap magnitude in the fluctuation-dominated (higher-temperature) regime. This is quite obvious when looking at the 40K data, for example, where the (residual) coherence peak position is nearly unchanged from the lower temperature data. I also notice that the gap values presented are also seemingly inconsistent between 1D and 1E; the coherence peak separation at 5K (Fig. 1D, red dI/dV curve) appears to be close to $2\Delta=30$ meV, but it is reported as approximately 10 meV in Fig. 1E. Can the authors explain this discrepancy? In any case, as good practice the fits used to generate the values in Fig 1E should be shown in the supplemental material (for all temperatures).

Reply (R2-6): We thank the reviewer for this insightful comment. We agree with the reviewer's opinion that the Dynes fitting can't give the exact results for the gap magnitude in the fluctuation-dominated regime. However, it is still a useful method to obtain the gap vanishing temperature, which has been used in Pb, Sn, FeSe films and so on previously (*Nat. Phys.* **6**, 104-108 (2010), *Sci. Bull.* **63**, 1332-1337 (2018), *Nano Lett.* **22**, 3245-3251 (2022)). Moreover, we also extract the zero bias conductance (ZBC) from STS, which shows a linear dependence on temperature near T_c^{onset} and gives a consistent $T_c^{\text{onset}} \sim 51$ K, as shown in Fig. S3c of the revised supplementary information. Besides, nearly all the temperature-dependent STS data with the gap magnitude ~ 15 meV give similar results, such as another dataset that we show in the

reply (R3-3). Therefore, we think the gap closing temperature derived from the fitting is reliable.

Regarding the gap discrepancy in Fig. 1D and Fig. 1E, generally the Dynes fitting gives a gap value that is smaller than directly read from STS due to the broadening Γ (*Phys. Rev. Lett.* **41**, 1509 (1978), *Nat. Phys.* **6**, 104-108 (2010), *Sci. Bull.* **63**, 1332-1337 (2018)). In our case, we use a simple S-wave gap function to fit the STS which gives rise to the gap discrepancy. Besides, Dynes fitting using an anisotropic S-wave gap function (*Phys. Rev. Lett.* **124**, 097001 (2020), *Phys. Rev. Lett.* **117**, 117001 (2016), *Nano Lett.* **22**, 3245-3251 (2022)) gives a similar gap magnitude with that directly read from STS, as shown in Fig.R1c-f in the reply (R3-3). Even though different gap function gives different gap magnitude, the gap vanishing temperature is consistent, as shown in Fig.R1.

Besides, with increasing temperature, the spectra become broadened and the coherence peaks are gradually suppressed with quasiparticles excited within the gap, but the peak position doesn't change obviously. The reduced gap value at high temperature is due to the increased broadening Γ , which is related with the lifetime of quasiparticles.

Following this valuable comment, we have added the raw data and the fittings of STS for all temperatures in Fig. S3 of the revised supplementary information.

Reviewer #3 (Remarks):

The superconductivity of one unit cell FeSe has been a hot topic in recent years. Although there are a lot of ex-situ and in-situ transport measurements on FeSe films, consensus on its superconducting T_c has not been reached. In this manuscript, Zhao et al. combined STM/STS and in situ micro-scale transport measurements to study the one-unit-cell FeSe grown on SrTiO₃ substrate. Two main conclusions are as follows:

1. The zero resistance temperature is consistent with the coherence peak vanishing temperature, and the superconducting onset temperature is consistent with the gap closing temperature.
2. Two-step power law V-I dependence is observed. The authors explain this behavior as the percolation via a network of superconducting puddles with different superconducting gaps.

Reply (R3-1): We thank the reviewer very much for reviewing our manuscript. Following his/her constructive comments and suggestions, we have revised our manuscript accordingly.

Currently, I cannot make a definitive recommendation as I have several questions and comments that the authors need to address.

I. To date, the superconducting T_c of FeSe on SrTiO₃ is controversial, and the proposed electronic inhomogeneity and phase fluctuations in this work could be a critical origin. However, the current study does not do a good enough job of discussing the main

inconsistencies in previous literature.

1) This work is not the first combined work of STM and micro-scale electrical transport (the way of describing ref.25 and the first sentence of the summary should be revised). An early work, with some of the authors being the coauthors, has used quite similar techniques of 4PP to measure 1uc FeSe films, but the measured zero resistance temperature is above 100K (Ref.25). It's a bit confusing to the readers why the T_c varies so much on 1uc FeSe measured with similar experimental techniques. The width between adjacent probes is even smaller in the current work(5 micrometers) than that in ref.25 (100 micrometers or 10 micrometers), which can reduce the effect of inhomogeneity. However, the measured T_c is much lower in this work, quite unexpected if electronic inhomogeneity is the main cause for the broadened superconducting transition. The discrepancy should be explicitly discussed.

2) Previous ARPES studies suggest pairing temperature around 65K. An early temperature-dependent STS work by the authors also suggests a pairing temperature of 68K (Zhang et al., Phy. Rev. B 89, 060506(2014)). Here the authors show the pairing temperature of 1uc FeSe is only 52K, much lower than reported, but consistent with the onset T_c . Is the reduced pairing temperature originated from the poorer sample quality? Do the authors expect the onset T_c of 68K on the optimized samples? In-situ transport measurements on the optimized samples with ~ 20 meV gap and two gap features should be shown. At least a discussion on the expected onset T_c for the optimized sample should be added.

Reply (R3-2): We thank the reviewer for the insightful comments.

1) We apologize for the inappropriate description. Here, “the first time” suggests we have, for the first time, combined STS and *in situ* transport. However, as pointed out by the reviewer, this could induce some misunderstanding and we have deleted “for the first time” in the first sentence of the summary in the revised manuscript. Regarding ref. 25, we doubt the measurement was probably influenced by the structural transition of SrTiO₃ substrate at 105 K. So we are very sorry that we couldn't give an explicit discussion between our results and ref. 25 from physical aspect at present.

2) In the previous version of the manuscript, we haven't well introduced the difference of FeSe films between those with line defects and those with twin boundaries. As we have added in the introduction on Page 3 of the revised manuscript and in the Fig. S1 of the revised supplementary information, there are two types of 1 UC FeSe films. The FeSe film with dense line defects has a nearly Gaussian-distributed gap distribution with the gap ranging from 13 to 20 meV (Fig. S1b), while the FeSe film with bright twin boundaries has two characteristic superconducting gaps, i.e. 15 meV and 10 meV, characterized by a bimodal Gaussian distribution as shown in Fig. S1d. A general consensus on superconducting gap (~ 20 meV) and pairing temperature (~ 68 K) is based on the former. If we assume the same superconducting mechanism holds for FeSe films with both types of boundaries and use the ratio of $2\Delta/k_B T_c$ to roughly estimate them, $20\text{meV}/68\text{K}$ (6.82) is almost consistent with $15\text{meV}/52\text{K}$ (6.69).

Therefore, we believe the reduced pairing temperature is the intrinsic nature of FeSe

films with twin boundaries. For the last several years, we have grown over several hundreds of 1 UC FeSe, and so far the best T_c^{zero} we have achieved is around 31 K. The T_c^{zero} exceeding 30 K could demonstrate the good quality of our samples. Moreover, a higher T_c^{onset} is hopeful in FeSe films with line defects boundaries and stronger interface coupling, which is also the next research topic we plan to study. This brings a challenge to the growth because it is very hard to grow FeSe films with larger domain diameter and uniform 20 meV gap. In fact, in experiments, the 20 meV gap is only restricted to certain location on the film.

Following this constructive suggestion, we have added “Note that under strong interface coupling, one can speculate a high T_c^{onset} as well as the pairing temperature in 1 UC FeSe with the superconducting gap ~ 20 meV, which deserves further *in-situ* investigations” on Page 5 of the revised manuscript.

II. The correspondence between the gap evolution and transport behavior cannot be directly observed from the raw data.

1) A third-order polynomial background is removed to analyze the gap size, while previous STS work does not show such a background. The authors should explain the origin of the background, its evolution with temperature, and how it would affect the gap determination.

2) According to the data in Fig.1d, the gap size seems constant below 45K, and the peak position always locates at about 15meV. The gap size of 45K seems even slightly larger than that of 40K. This is in contrast to the fitted gap size, which decreases with increasing temperature. The fitting using the Dynes function deviates from the data (Fig.S2). Is the poor fitting induced by the gap variation with temperature? The authors should show the fitted results at different temperatures.

3) The coherence peak can be hardly observed from the raw data (Fig.S2). Is it because of the poor sample quality? The definition of T_{cp} of 30K is based on the vanished coherence peak. However, one can still see a peak at around 15meV in the 32K or even 40K data. The authors need to explain how they define the coherence peak, so the readers can judge the existence by themselves. Besides, the peak structure might be related to the background subtraction. Can the author show the raw data and are there dramatic changes around 30K in the raw data?

Reply (R3-3): We thank the reviewer for the insightful comments.

1) We think that the background is probably due to the high Nb doping of SrTiO₃ substrates. For different batches of substrates, the background is always different. As shown in Fig.2 of *Chin. Phys. Lett.* **31**, 017401 (2014), the STS background of 1 UC FeSe on Nb doped SrTiO₃ substrate (Fig. 2d) is higher than that on insulating SrTiO₃ substrate (Fig. 2c). Also, the high background can often be found in 1 UC FeSe grown on Nb doped SrTiO₃ from Shinkosha company (*Phys. Rev. B* **98**, 121410 (2018), *Phys. Rev. B* **101**, 205421 (2020)). The STS background can be normalized by polynomial fittings (*Phys. Rev. B* **98**, 121410 (2018), *J. Electron Spectrosc.* **109**, 147-155 (2000), *Nano Lett.* **19**, 3464-3472 (2019)). We use the same fitting method as these previous

studies.

To demonstrate the repeatability and reliability of the gap closing temperature, we show the raw data and the fitting results of another dataset of STS in Fig. R1, which give a consistent gap vanishing temperature. Moreover, using higher order polynomial fitting gives the consistent gap closing temperature as shown in Fig. R2.

Fig. R1 | Fitting results of another dataset of dI/dV spectra. **a**, The original dI/dV spectra at different temperatures. **b**, The temperature dependence of the ZBC extracted from the dI/dV spectra in **a**, giving a $T_p \sim 52$ K. **c**, Normalized spectra (open symbols) and BCS fittings (solid curves) at each temperature, by using the Dynes fitting with S-wave gap function. **d**, The fitted gap magnitude of the dI/dV spectra in **c**, giving a $T_p \sim 53$ K. **e**, Normalized spectra (open symbols) and BCS fittings (solid curves) at each temperature, by using the Dynes fitting with anisotropic S-wave gap function. **f**, The fitted gap magnitude of the dI/dV spectra in **e**, giving a $T_p \sim 57$ K.

Fig. R2 | Fitting results of dI/dV spectra in Fig. S3a with **a-b**, fourth-order, **c-d**, fifth-order, **e-f**, sixth-order polynomial background normalized.

Given that the gap at 50 K is almost vanished as shown in Figs. S3a and S3b, it is reasonable to expect a gap closing temperature just above 50 K.

In order for the readers to better understand its evolution with temperature, we present the raw data in Fig. S3a of the revised supplementary information.

2) With increasing temperature, the gaps fill up, that is, coherence peaks gradually reduce but maintain at 15 meV until disappear. At high temperatures the broadening Γ (related with the lifetime of quasiparticles) (*Phys. Rev. Lett.* **41**, 1509 (1978)) in Dynes function is larger, and that is why the fitting gives a smaller gap value eventually. Regarding the poor fitting in the original supplementary Fig.S2, the poor fitting here is because we employed a simple S-wave gap function during Dynes fitting and this gives a gap size smaller (~ 10 meV) than read directly from STS (~ 15 meV). However, if we turn to use an anisotropic S-wave gap function as demonstrated in literatures (*Phys. Rev. Lett.* **124**, 097001 (2020), *Phys. Rev. Lett.* **117**, 117001 (2016), *Nano Lett.* **22**, 3245-3251 (2022)), the gap size is around 15 meV as shown in Fig.R1c-f. Even though different gap function gives different gap magnitude, the gap vanishing temperature is consistent, as shown in Fig. R1.

Following this nice suggestion, we have added all the fittings for all temperatures in Fig. S3b of the revised supplementary information.

3) The main reason why the coherence peak is hardly observed from the raw data is that the high background obscures the peak, which is often found in FeSe films grown on certain substrates with high Nb doping (*Phys. Rev. B* **98**, 121410 (2018), *Phys. Rev. B* **101**, 205421 (2020)).

As the reviewer points out, the coherence peak might be related to the background subtraction. We define T_{cp} as the vanishing temperature of coherence peak, which can be obtained from the STS kink of the raw data. As shown in Fig. S3a of the revised supplementary information, the T_{cp} is about 32 K. No obvious kink can be observed from STS at higher temperatures.

Following this valuable suggestion, we have added “**Notably, the coherence peak of the superconducting gap ~ 15 meV vanishes at ~ 32 K (Fig. 1d and Supplementary Fig. S3a). We define this temperature as T_{cp} .**” on Page 5 of the revised manuscript for the readers to better understand its definition.

III. For the micro-scale transport measurement, one doubt about Fig. 3 is how to prove that the low current regime and the high current regime are from the contributions of domain boundaries and domains, respectively. In the manuscript, the author claimed it's supported by the existence of an $\alpha=0$ plateau, which is not clear. Why would the coexistence of superconducting regions and non-superconducting regions induce a $\log V \sim \text{constant}$ region in IV curve? Because this is the main conclusion of this manuscript, more data should be shown to support this claim. For example, comparing $V(I)$ relation on the sample surface with different portions of domain boundaries and homogeneous domains would be helpful. Since the data in Fig.3 is from a highly homogeneous sample as the authors claimed, results from a more inhomogeneous sample should be shown and compared with Fig.3 to provide stronger evidence.

Reply (R3-4): We thank the reviewer for the valuable comment. Firstly, we would like to apologize for the inappropriate description of $\alpha \sim 0$. We proposed the $\alpha \sim 0$ plateau

in order to emphasize the distinct divide between the low current regime and the high current regime in the $V(I)$ curves. Accordingly, we have deleted the description of $\alpha \sim 0$.

Besides, we claimed that the 1 UC FeSe film in this study had improved homogeneity, which was compared with FeSe films with dense line defects. As shown in Fig. 1, the FeSe film doesn't show lines of Fe vacancies or other obvious defects, which can suppress the superconductivity.

Moreover, we agree with the reviewer that comparing $V(I)$ relations on the sample surface with different portions of domain boundaries would be helpful. Accordingly, we conducted additional experiments to grow FeSe films with different portions of twin boundaries. However, it is very hard to quantitatively control the portions of twin boundaries from MBE growth, unless we try to introduce some line defects as shown in the inset of Fig. R3b. This is also an inhomogeneous sample as the reviewer suggests.

Fig. R3 | **a**, $V(I)$ characteristics on a double-logarithmic scale. **b**, Temperature evolution of the exponent α , extracted from the power-law fittings in regimes H and L in **a**, giving $T_H = 21.8$ K and $T_L = 20.8$ K when $\alpha = 3$. The inset shows the topographic image.

Considering the divergent α - T behavior in the high-current regime and the consistent $(\text{dln}R/\text{dT})^{-2/3}$ - T analysis in Fig. 3, the V/I surge in the high-current regime is probably caused by the proliferation of free topological vortices, which is the celebrated BKT transition, dominated by FeSe in domains. Given the distinct two superconducting phases feature in 1 UC FeSe with twin boundaries, we think the V/I increase in the low-current regime is likely due to the weak superconductivity of FeSe film on twin boundaries (compared with those in domains). Another possible explanation for the low-current regime is that there might be some vortex movement due to the intricacies of the two-fluid like structure (FeSe in domain and on twin boundaries) under driving current.

A previous study (*Phys. Rev. Lett.* **109**, 137004 (2012)) suggests twin boundaries tend

to pin vortices in FeSe films grown on graphitized SiC substrate, which reduces the possibility of vortex movement. As shown in Fig. R3, by introducing line defects to further restrain the vortex movement (*2D Mater.* 6, 021005 (2019)), we can still observe the clear two-step $V(I)$ feature, demonstrating the most likely possibility is due to the weak superconductivity with weak superfluid phase stiffness on boundaries. However, the system becomes more complex, as the portion of twin boundaries, the BKT transition temperature and gap distribution have all changed. It's difficult to quantitatively analyze the results.

Even though we can't give an explicit physical picture at present, the unique two-step $V(I)$ behavior helps to understand the spectroscopic and spatial two-gap feature in 1 UC FeSe and may invoke more related investigation.

IV. The V-I curve in Fig.2d seems less smooth than the V-I curve in the double-logarithmic scale in Fig.3a. Are the data from different datasets or differently treated? Are they consistent if they are from different datasets?

Reply (R3-5): They are from different datasets. But we only measured to 0.6 mA for the dataset of Fig. 2d. Judging from 0 to 0.6 mA, the two datasets are consistent.

REVIEWER COMMENTS

Reviewer #1 (Remarks to the Author):

The authors have addressed all the points I raised in the last review report.

Reviewer #2 (Remarks to the Author):

The edits made by the authors have, in my opinion, improved the manuscript overall. In particular, I find that the discussion and data surrounding the core observations of nano-scale gap behavior is much clearer than in the initial submission. I appreciate that the authors have taken considerable effort to add additional material addressing my initial comments, most of which I found adequate.

Considering this, I am inclined to agree with the authors that this combined STM/STS + in situ transport study, by providing comparative data from both techniques on the same films to allow for consistent comparisons [between nanoscale gap structure and macroscopic transport behavior], represents a sufficiently novel result to, in principle, warrant publication in Nature Communications.

However, I think that this study still has some important flaws and omissions which hamper it, and that the manuscript should not be published until they are satisfactorily addressed.

First, I still have qualms with the manuscript's gap analysis as presented. As the authors admit in their rebuttal, an apparent reduced gap value from this Dynes fitting approach can often be convoluted with an decrease in the quasiparticle lifetimes (corresponding to an increase in the broadening Γ , presumably a fitting parameter in their gap fits). This can also be true when comparing extracted gap spectra on different regions of the same sample. The authors should demonstrate more thoroughly that the behavior they measure in Figures 1F + 1G comparing the gap behavior in the domain and boundary regions cannot be explained by a systematic difference in Γ , rather than Δ (showing Γ from the fits would be helpful). As it stands, this is unclear – looking at 1F and 1G, the extracted gap values at low temperatures are quite similar despite major differences in the apparent quasiparticle peak separation, so it seems reasonable to assume that Γ is the dominant difference. I don't believe that this is a nitpicky complaint, because the difference between the twin domain boundary regions having a suppressed Δ versus a greatly enhanced Γ substantively changes the physical interpretation presented by the authors in Figure 4(b-d) of electronic versus structural/disorder inhomogeneity within the boundary regions.

Second, I believe the strength and novelty of this combined approach really lies in the ability to compare nanoscale gap phenomena and superconducting behavior across separate films/regions with different transport behavior, most notably different T_{c0} . It is clear from the authors' response to Reviewer 3 ("For the last several years, we have grown over several hundreds of 1 UC FeSe", and the data in Fig. R3) that they have performed equivalent measurements on additional films with lower T_{BKT} (and presumably reduced T_{c0}). Can the authors show correlations between the resistive transition behavior and nanoscale gap behavior on such other films with reduced T_{c0} ? Do films with reduced T_{c0} show systematically higher twin domain boundary densities, or is T_{c0} better correlated with other factors such as disorder within the domain regions (which presumably dominate the transport behavior)? If the former, I believe that this would provide much more definitive evidence for their important and novel conclusion that "the electronic inhomogeneity around the boundaries limits T_{BKT} in 1 UC FeSe". This important comparison should be included in the main text.

Reviewer #3 (Remarks to the Author):

I appreciate the authors' efforts in addressing the Reviewers' comments, and I find them satisfying in most cases.

The following issues remain to be addressed:

1. As there have already been several related literatures about the in-situ transport and ex-situ transport measurements on FeSe/STO, clear discussions on the differences and similarities between the data obtained here and those previously reported are required, such as the comparison between $T_c(\text{onset})$, T_{p1} with those measured by previous transport and temperature-dependent STS/ARPES measurements. Some discussions on the possible origins that cause these differences are needed as well. Such information would benefit the readers and the community.
2. On page 4, the authors added some discussions on the lattice anisotropy in the adjacent two domains. First, it is hard to see $a_0 > b_0$ from the figure. Second, if the anisotropy is actually there, is there an explanation on why the structure is orthorhombic in 1uc FeSe/STO? 1uc FeSe is heavily electron doped and nematic order is not there. The strain from the STO substrates would also gives $a=b$. The driving force for $a \neq b$ is not clear.
3. The inclusion of error bars is imperative for parameters derived from the fitting results in both the main text and supplementary information figures. Parameters such as gap magnitude, gap height, and exponent α , should be accompanied by error bars for enhanced precision and clarity.

With the proper corrections concerning the points mentioned above, I am happy to recommend this manuscript for publication in Nature Communications.

Dear reviewers,

We sincerely thank all the reviewers for kindly handling our manuscript and providing scientific, rigorous and insightful comments and suggestions. We have considered all the comments and suggestions carefully and revised the manuscript accordingly. The responses to the reviewers' comments are listed as follows. The major modifications in the main text are highlighted in red color.

=====Authors' responses to the reviewers' comments=====

Reviewer #1 (Remarks):

The authors have addressed all the points I raised in the last review report.

Reply: We thank the reviewer very much for the valuable comments and suggestions in the last review report, which have significantly improved the quality of our manuscript.

Reviewer #2 (Remarks):

The edits made by the authors have, in my opinion, improved the manuscript overall. In particular, I find that the discussion and data surrounding the core observations of nano-scale gap behavior is much clearer than in the initial submission. I appreciate that the authors have taken considerable effort to add additional material addressing my initial comments, most of which I found adequate.

Reply: We are very glad for the reviewer's recognition of our revised manuscript. We also appreciate the reviewer's warm work earnestly for further reviewing our manuscript and we have revised the manuscript according to his/her suggestions.

Considering this, I am inclined to agree with the authors that this combined STM/STS + in situ transport study, by providing comparative data from both techniques on the same films to allow for consistent comparisons [between nanoscale gap structure and macroscopic transport behavior], represents a sufficiently novel result to, in principle, warrant publication in Nature Communications.

Reply: We thank the reviewer very much for recognizing the novelty of our work.

However, I think that this study still has some important flaws and omissions which hamper it, and that the manuscript should not be published until they are satisfactorily addressed.

Reply: We thank the reviewer for the valuable comments and suggestions and we learn

a lot from them.

First, I still have qualms with the manuscript's gap analysis as presented. As the authors admit in their rebuttal, an apparent reduced gap value from this Dynes fitting approach can often be convoluted with an decrease in the quasiparticle lifetimes (corresponding to an increase in the broadening Γ , presumably a fitting parameter in their gap fits). This can also be true when comparing extracted gap spectra on different regions of the same sample. The authors should demonstrate more thoroughly that the behavior they measure in Figures 1F + 1G comparing the gap behavior in the domain and boundary regions cannot be explained by a systematic difference in Γ , rather than Δ (showing Γ from the fits would be helpful). As it stands, this is unclear – looking at 1F and 1G, the extracted gap values at low temperatures are quite similar despite major differences in the apparent quasiparticle peak separation, so it seems reasonable to assume that Γ is the dominant difference. I don't believe that this is a nitpicky complaint, because the difference between the twin domain boundary regions having a suppressed Δ versus a greatly enhanced Γ substantively changes the physical interpretation presented by the authors in Figure 4(b-d) of electronic versus structural/disorder inhomogeneity within the boundary regions.

Reply: We thank the reviewer for the nice suggestion and we have added the parameter Γ in Tables S1 and S2 of the revised supplementary information accordingly to depict the fittings more clearly. The Γ , which is the spectral broadening relating with the quasiparticle lifetime, is an important parameter given by the fittings. As shown in Table S1, even though Γ is similar from 38 K to 50 K fittings, the gap value decreases monotonically with increasing temperature, which demonstrates our results can't be explained by a systematic difference in Γ but reflect the true gap features. Besides, Tables S1 and S2 don't show big difference in Γ used in Figs. 1f and 1g, guaranteeing the physical interpretation in Figs. 4b-d. As has been pointed out in the first response letter, the deviation of gap value from the coherence peak position in the raw data is because we choose to use a relatively simple single S-wave Dynes fitting while the true superconducting gap of monolayer FeSe on SrTiO₃ is anisotropic in momentum space (*Phys. Rev. Lett.* **124**, 097001 (2020), *Phys. Rev. Lett.* **117**, 117001 (2016)). Since we only care about the gap closing temperature in this study, single S-wave Dynes fitting is enough in this case (as we have shown in the first response letter, using anisotropic S-wave fitting or single S-wave fitting gives the similar gap closing temperature).

Given that the gap is almost vanished at 50 K in domains and has already vanished at 41 K on boundaries, as shown in Figs. S3a-b and S3e-f, it is reasonable to expect a gap closing temperature just above 50 K in domains and below 41 K on boundaries. Besides, the linear fittings of gap height and zero bias conductance give consistent results. Therefore, we believe the Dynes fitting itself doesn't influence the main findings and the conclusion of this manuscript.

Second, I believe the strength and novelty of this combined approach really lies in the ability to compare nanoscale gap phenomena and superconducting behavior across

separate films/regions with different transport behavior, most notably different T_{czero} . It is clear from the authors' response to Reviewer 3 ("For the last several years, we have grown over several hundreds of 1 UC FeSe", and the data in Fig. R3) that they have performed equivalent measurements on additional films with lower T_{BKT} (and presumably reduced T_{czero}). Can the authors show correlations between the resistive transition behavior and nanoscale gap behavior on such other films with reduced T_{czero} ? Do films with reduced T_{czero} show systematically higher twin domain boundary densities, or is T_{czero} better correlated with other factors such as disorder within the domain regions (which presumably dominate the transport behavior)? If the former, I believe that this would provide much more definitive evidence for their important and novel conclusion that "the electronic inhomogeneity around the boundaries limits T_{BKT} in 1 UC FeSe". This important comparison should be included in the main text.

Reply: We thank the reviewer for the nice suggestion. Firstly, the content of twin boundaries is small and similar among the samples we measured and it is hard to quantitatively control the portions of twin boundaries from MBE growth at least on the certain batches of substrates we used. Secondly, we find that T_c^{zero} is consistent with $T_{BKT} \sim T_H$, which is dominated by the domains, rather than boundaries. In our experiments, we find two common situations with lower T_c^{zero} .

1. If the Se/Fe ratio is larger or the post-growth annealing is not long enough, T_c^{zero} and T_c^{onset} , as well as the superconducting gap, decrease simultaneously. A typical correlation between the resistive transition and the gap behavior is shown below. The STM topography and the content of twin boundaries don't show obvious change.

Fig. R1 | **a**, STM topographic image of 1 UC FeSe with lower T_c^{zero} . **b**, Temperature-dependent dI/dV spectra taken in domains, showing the gap closing temperature ~ 42 K. **c**, Temperature-dependent resistance, showing $T_c^{zero} \sim 23.3$ K and $T_c^{onset} \sim 43.0$ K. The inset gives $T_{BKT} \sim 23.0$ K.

2. If the FeSe films are prepared with a substantial amount of defects and impurities, T_c^{zero} also decreases. Some samples show lower T_c^{zero} and some samples are even not superconducting, which depend on the sample quality. Typical R - T curves are

shown in Fig. R2. In this case, there are no clear correlations between the resistive transition and the gap behavior, as it is inhomogeneous and the gap behavior differs in different locations.

Fig. R2 | **a**, STM topographic image of 1 UC FeSe with impurities. **b**, Typical temperature-dependent resistance curves.

The above two situations show no more information beyond the main text. Therefore, we didn't add them in the main text. As for the two factors (domain boundary densities and disorder) the reviewer raised which reduce T_c^{zero} , we are sorry that we don't have experimental evidence of correlations between T_c^{zero} and twin boundary densities at present. Besides, it is understandable that T_c^{zero} decreases as disorders induced within the domains.

Regarding the sentence “the electronic inhomogeneity around the boundaries limits T_{BKT} in 1 UC FeSe”, we think that the phase fluctuation (*Phys. Rev. X* **11**, 021054 (2021)) and spatial inhomogeneity (*Phys. Rev. B* **108**, 214514 (2023)) are two key factors impacting $T_{\text{BKT}} \sim T_c^{\text{zero}}$, as disorder can influence the vortex structures of 2D superconductors and decrease the superfluid phase stiffness J drastically (*Phys. Rev. B* **80**, 214506 (2009), *J. Phys.: Condens. Matter* **34**, 083001 (2022), *Rev. Mod. Phys.* **66**, 1125 (1994), *2D Mater.* **6**, 021005 (2019)). Moreover, by comparing our results with a recent study (*Phys. Rev. Lett.* **125**, 097003 (2020)), the influence of electronic inhomogeneity resulting from domain boundaries on T_c^{zero} is more obvious. This study shows a similar superconducting transition in an organic ion intercalated superconductor $(\text{TBA})_x\text{FeSe}$, where the distance between adjacent FeSe layers is enlarged from ~ 5.5 Å in pristine FeSe to 15.5 Å by TBA^+ intercalation. Regardless of the strong phase fluctuation arising from the two-dimensionality in this system, it exhibits a sharper superconducting transition than 1 UC FeSe/SrTiO₃. The onset temperature where the resistance starts to decrease is ~ 55 K, and T_c^{zero} ($\sim T_{\text{BKT}}$) is ~ 43 K. The prominent difference between this system and 1 UC FeSe/SrTiO₃ is the lack of domains, which indicates that the electronic inhomogeneity around domain boundaries

in 1 UC FeSe/SrTiO₃ contributes to the broadening of superconducting transition.

Following this valuable suggestion, we have added five references (*J. Phys.: Condens. Matter* **34**, 083001 (2022), *Phys. Rev. B* **80**, 214506 (2009), *Rev. Mod. Phys.* **66**, 1125-1388 (1994), *2D Mater.* **6**, 021005 (2019), *Phys. Rev. B* **108**, 214514 (2023)) on Page 8 of the revised manuscript for the readers to better understand this sentence.

Reviewer #3 (Remarks):

I appreciate the authors' efforts in addressing the Reviewers' comments, and I find them satisfying in most cases.

The following issues remain to be addressed:

Reply: We thank the reviewer very much for further reviewing our manuscript. Following his/her nice comments and suggestions, we have revised our manuscript accordingly.

1. As there have already been several related literatures about the in-situ transport and ex-situ transport measurements on FeSe/STO, clear discussions on the differences and similarities between the data obtained here and those previously reported are required, such as the comparison between $T_c(\text{onset})$, T_{p1} with those measured by previous transport and temperature-dependent STS/ARPES measurements. Some discussions on the possible origins that cause these differences are needed as well. Such information would benefit the readers and the community.

Reply: We thank the reviewer for the nice suggestion and we agree that the comparison with previous results would be beneficial, which we also meant to do before. However, we find it is difficult to make meaningful comparisons because there are many discrepancies on T_c^{zero} and T_c^{onset} among previous studies even with the same measurement technique. For instance, ARPES measurements from different groups reported T_c^{onset} ranging from 50 K to 83 K. This is probably because different sample quality and status gives different results, which brings the long-standing puzzle on the superconducting transition temperature of FeSe/SrTiO₃ and also shows the importance of our measurement technique combining STM/STS and *in-situ* microscale transport.

Following this valuable suggestion, we have added a comparison of T_c measurements across different techniques and related discussions in Table S3 on Page 7 of the revised supplementary information.

2. On page 4, the authors added some discussions on the lattice anisotropy in the adjacent two domains. First, it is hard to see $a_0 > b_0$ from the figure. Second, if the anisotropy is actually there, is there an explanation on why the structure is orthorhombic in 1uc FeSe/STO? 1uc FeSe is heavily electron doped and nematic order is not there. The strain from the STO substrates would also gives $a = b$. The driving force for $a \neq b$ is

not clear.

Reply: We thank the reviewer for the valuable comment. In our experiment, it is for sure that the lattice anisotropy exists in the adjacent two domains. Besides, the twin boundary and the lattice anisotropy also exist in bulk FeSe, FeSe films grown on graphene and other iron pnictides (*Phys. Rev. X* **5**, 031022 (2015), *Phys. Rev. Lett.* **109**, 137004 (2012), *Science* **327**, 181 (2010), *Nat. Phys.* **10**, 225 (2014)). In these systems, the driving force is the tetragonal-to-orthorhombic structural phase transition. However, as pointed out by the reviewer, 1 UC FeSe on SrTiO₃ is heavily electron doped and doesn't show nematic order. To our knowledge, no firm explanation for $a_0 > b_0$ in FeSe/SrTiO₃ has been put forward in previous literatures. We think it is likely due to the strain from the SrTiO₃ substrates. Firstly, the in-plane lattice constant of FeSe (0.378 nm) is smaller than SrTiO₃ (0.39 nm). Secondly, SrTiO₃ undergoes a cubic-to-tetragonal structural phase transition at ~105 K, resulting in $a=b \neq c$. But the surface of the SrTiO₃ substrates could be ac or bc planes, giving anisotropic strain to 1 UC FeSe. Thirdly, accompanied by the phase transition, SrTiO₃ forms a dense network of domains and boundaries (*Phys. Rev.* **134**, A981 (1964)). Therefore, the relaxation of strain between SrTiO₃ and FeSe could possibly induce the lattice anisotropy in 1 UC FeSe. More experiments are still needed to clarify this question.

3. The inclusion of error bars is imperative for parameters derived from the fitting results in both the main text and supplementary information figures. Parameters such as gap magnitude, gap height, and exponent α , should be accompanied by error bars for enhanced precision and clarity.

Reply: We thank the reviewer for the nice suggestion. We have added error bars in the related figures (Figs. 1-3 and S3) accordingly.

With the proper corrections concerning the points mentioned above, I am happy to recommend this manuscript for publication in Nature Communications.

Reply: We sincerely thank the reviewer for the professional review work on our manuscript and hope that the corrections could meet with approval.

REVIEWERS' COMMENTS

Reviewer #2 (Remarks to the Author):

The authors' most recent reply and edits to the manuscript reasonably address the majority of my previous concerns. In particular, the comments and additional data (Figs. R1 and R2) shared in the most recent reply regarding the relative importance of domains versus boundaries in determining T_{c0} (or semi-equivalently TBKT) are useful and enlightening. The comparison to bulk (TBA+)FeSe crystals is a good one, and I agree that it is highly suggestive. In my opinion the manuscript would be improved if some of this important context were included in the manuscript directly (rather than embedded in the added references in the discussion section), to more explicitly clarify their argument for which of the domains versus boundary regions sets T_{c0} /TBKT.

Following this suggestion, I would be happy to recommend publication in Nature Communications.

Reviewer #3 (Remarks to the Author):

The authors have responded satisfactorily to my comments and have revised the manuscript accordingly. I recommend its publication in Nature Communications.

Dear reviewers,

We would like to thank the reviewers for taking their time to review our manuscript and for their valuable comments for improving our paper. We have considered the remaining comments and suggestions carefully and revised the manuscript accordingly.

=====Authors' responses to the reviewers' comments=====

Reviewer #2 (Remarks):

The authors' most recent reply and edits to the manuscript reasonably address the majority of my previous concerns. In particular, the comments and additional data (Figs. R1 and R2) shared in the most recent reply regarding the relative importance of domains versus boundaries in determining T_{c0} (or semi-equivalently TBKT) are useful and enlightening. The comparison to bulk $(TBA^+)FeSe$ crystals is a good one, and I agree that it is highly suggestive. In my opinion the manuscript would be improved if some of this important context were included in the manuscript directly (rather than embedded in the added references in the discussion section), to more explicitly clarify their argument for which of the domains versus boundary regions sets $T_{c0}/TBKT$. Following this suggestion, I would be happy to recommend publication in Nature Communications.

Reply: We sincerely thank the reviewer for the positive comments and suggestions and the recommendation for publication of our manuscript. Following this valuable suggestion, we have added the comparison with $(TBA)_xFeSe$ single crystal and related discussions in the manuscript.

Reviewer #3 (Remarks):

The authors have responded satisfactorily to my comments and have revised the manuscript accordingly. I recommend its publication in Nature Communications.

Reply: We sincerely thank the reviewer for the professional review work on our manuscript, which has helped to take this work to a new level.